# Improved Active and Reactive Control of a Small Wind Turbine System Connected to the Grid

**Theofilos Papadopoulos \*, Emmanuel Tatakis \* and Efthymios Koukoulis**

Department of Electrical & Computer Engineering; University of Patras, Rion-Patras 265 04, Greece; ece7814@upnet.gr

\* Correspondence: teopap@mail.ntua.gr (T.P.); e.c.tatakis@ece.upatras.gr (E.T.)

**Abstract:** This paper deals with the interconnection of a small wind turbine with the low voltage distribution grid and the implementation of an improved control scheme, which also serves educational purposes. Initially the subsystems—wind turbine, rectifying bridge, interleaved boost converter, three-phase inverter, interconnection inductors, lifting transformer, filtering capacitors—are investigated, in order to explain their selection, based on the LEMEC (Laboratory of Electromechanical Energy Conversion, Department of Electrical Engineering, UoP) educational policy. Afterwards, the three-phase inverter control scheme, which is responsible for controlling its input voltage (voltage of the DC Bus) and consequently the active power, as well as the reactive power injected into the grid (VQ control) is analyzed. This is accomplished through DQ transformation and PI controllers which are responsible for generating the appropriate reference signals, to generate the required Space Vector Pulse Width Modulation (SVPWM) pulses to drive the semiconductor switches of the inverter. In addition, it is explained how this particular control method can compensate reactive power in the grid, even in apnea, by automatically charging the DC Bus. Finally, simulation and experimental results are given to prove the proposed control method effectiveness.

**Keywords:** wind turbine system; PQ control; VQ control; reactive power compensation; educational systems

---

## 1. Introduction

Problems caused by the use of fossil fuels [1–4], although known for decades, have begun to occupy much of the public debate on the sustainability of the production model in recent years. The immediate consequence of this, is the orientation of states and industry to the invention and the exploitation of new technologies [5], which will have less or no environmental impact, but can at the same time compete with the thermal stations and the internal combustion engines. The two main sectors, that is, those who are responsible for the overwhelming majority of greenhouse gases, are the electricity production and transportation [6].

Based on the broader strategy, a great deal of research is being given to the development of electronic power converters, which aim to control the driving and power generation systems in the best possible way [7,8]. In this framework, a system connecting a small wind turbine with the low voltage distribution grid, has been analyzed, simulated and implemented. The proposed improved control scheme can find the point of maximum power transfer (MPPT), stabilize the voltage of the DC Bus, which interconnects the boost converter with the three-phase inverter, by injecting into the grid the appropriate amount of active power and can also inject reactive power to the grid, in order to compensate it (VQ Control). The latter has a twofold purpose: to reduce transmission line losses, since the reactive current is provided locally by the RES and help with the stabilization of the grid in the

event of a fault, a tactic that has been promoted in recent years, with the development of microgrids [9] and the penetration of RES into the main grid.

The general idea of this topology can be seen in Figure 1, where the energy harvested from the wind turbine is transferred to the grid, via a boost converter and an inverter. Additionally, it is possible to implement a Storage Unit (SU), that is, a battery pack, the charging of which can be achieved with the SU Power Supply (SUPS). In any case, the three-phase inverter undertakes the energy transfer, stored in the SU or supplied directly from the SUPS, to the grid. This means that the system can also act as an Uninterruptible Power Supply (UPS). In case this function is not needed, the switch of the SU opens, so wind turbine feeds its power directly to the grid. Depending on SU switch state, different control methods can be used (PQ, VQ, Pf etc.), but in this paper, due to the lack of a SU, only the VQ control method is presented.

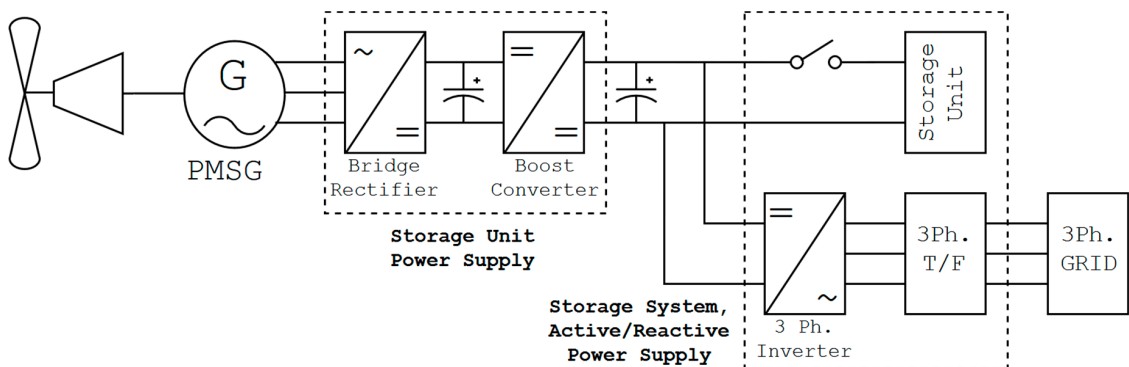

**Figure 1.** General topology of small Wind Turbine (WT) interconnection system, with UPS utilization.

This paper is based on the work done by the authors [10] at LEMEC and in previous work done in the same laboratory [11–13], considering inverters for small residential wind turbines and interconnection with the low voltage grid. It also takes into account textbooks and the work of other articles that provide some theoretical background on voltage source inverters (VSIs) and current source inverters (CSIs) [14–17], which are utilized in order to interconnect renewable energy sources (RES) with the grid. In every case, an inductor stands between the inverter and the grid. The voltage difference—amplitude and phase—between its terminals, determines the active and reactive power magnitude and flow. In order to filter current's harmonics, capacitors are used on the inverter or grid side, since the current's total harmonics distortion (THD) must not exceed a certain value—typically 5%.

Furthermore, this paper considers control architectures and methods from other papers [18–24], in order to combine the above and produce a simple, stable and reliable interconnection system, which is suitable both for real-world use and demonstration purposes. A large portion of articles deals with the decoupling of active and reactive power, so they can be controlled as two separate and independent variables [25,26]. This paper presents another version of this decoupling principle, as shown in Section 3, based on the mathematical and physical model that is derived. One more important point is that the most appropriate control methods use power, current and voltage variables as references, all of which are fed into PI controllers. This is done in order to apply a strict range of possible values, hence protect the system from instability and saturation faults. In the process of control design, an improved fine-tuning method was developed in order to find the PI parameters, which takes into consideration the non-linearities and the restriction for each state variable.

## 2. System Description

The system under study includes several devices, which do not guarantee optimization of its performance—since no storage unit was implemented, boost converter could be avoided. However, this topology provides the basis for in-depth analysis of every separate device and proves the concept

of operation. The complete circuit diagram is shown in Figure 2. The whole system includes, from left to right: a synchronous three-phase permanent magnet generator (PMSG), an uncontrolled 6-pulse three-phase bridge (FBR), an interleaved boost converter (IBC), a three phase inverter (INV), the DC Bus connecting the latter two, three inverter-grid interconnect coils (L$_I$), a three-phase transformer (TF) with a x13 lift ratio and three capacitors (C$_G$) for current harmonics filtering.

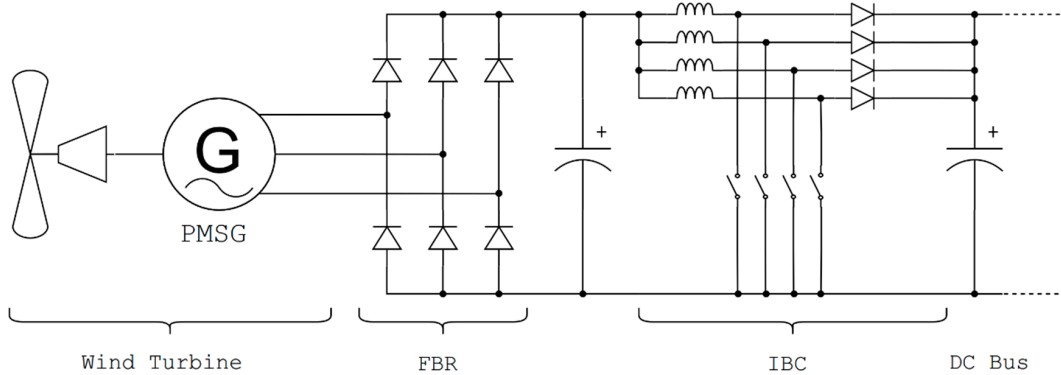

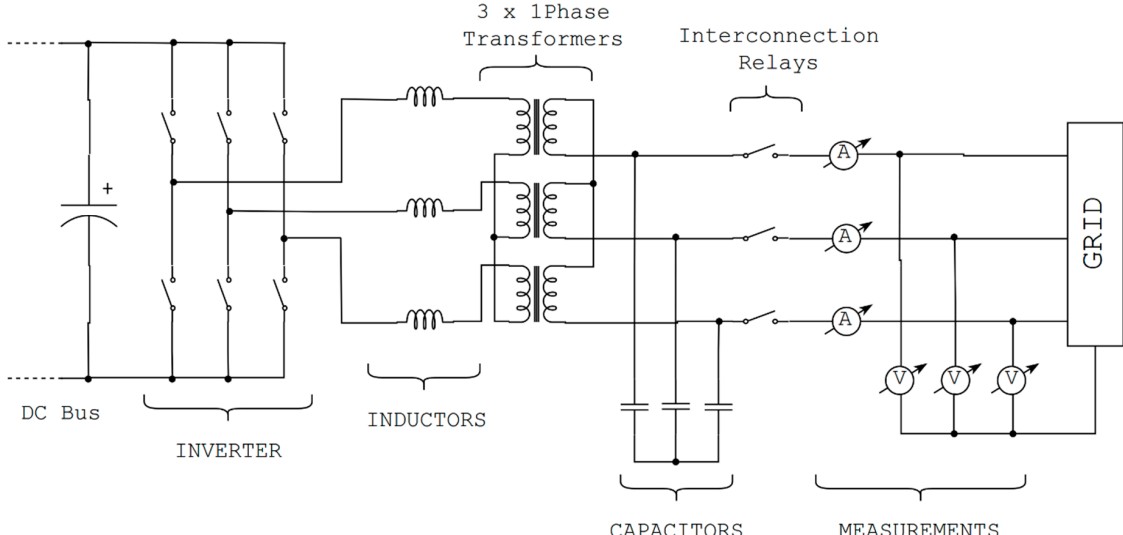

**Figure 2.** Simplified representation of the entire system.

Considering a sufficiently large DC Bus Capacitor, the overall system can be divided into two smaller ones, each one performing its own role. The first—PMSG, FBR, IBC—is aiming at maximizing energy exploitation, that is, wind energy potential. The second—INV, L$_I$, TF, C$_G$—aims to inject this energy to the grid by adjusting the harmonic content of the current and providing the desired reactive power injection. Objective of each device:

- PMSG: Power generation, with voltage and frequency proportional to the rotation speed of the shaft, that is, the wind speed. Minimum and maximum output voltage 40 and 120 V respectively. Nominal speed 1000 rpm, for wind speed 11.6 m/s.
- FBR: Uncontrolled three-phase 6-pulse bridge. It aims at rectifying and smoothing the output voltage of the wind turbine to provide a DC voltage, regardless of the wind speed.
- IBC: Finding the optimum point of operation of the wind turbine using MPPT methods. The use of an interleaved type converter enables more power at lower cost and a reduction in ripple.
- DC Bus: Separation of the two subsystems and intermediate energy storage to control active power supply to the grid. The total capacity is 10 mF.

- INV: Three-phase inverter with IGBT semiconductor switches (single module) and SVPWM pulsing method. It aims to convert the DC voltage at its input, to AC at the output and to control the active and reactive power injected to the grid, through the parameters m (modulation width) and δ (phase difference between the mains voltage and the basic harmonic voltage of the inverter).
- $L_I$: Three single phase inductors with 2 mH induction. A sufficiently large induction was chosen so that there are correspondingly large margins for adjusting the angle δ.
- TF: Three single phase transformers, with a 1:13 lifting ratio. In addition to lifting, they offer a galvanic isolation between the inverter and the grid, essential for the safe operation of the system in a laboratory environment.
- $C_G$: Three capacitors, which are aiming to suppress the higher harmonics of the current and are permanently connected to the side of the network, compensate part of the reactive power.

Both converters are controlled by separate control boards, which include a microcontroller and the necessary peripherals to drive the pulses of the semiconductor elements. It is therefore clear that although an interconnection system may consist of only the generator, the rectifier and the inverter, the present system offers the possibility of deepening into a plurality of elements and devices and combines four of the most important sectors of electrical engineering: power electronics, electrical and automation systems, electronics and microprocessor programming.

At this point, it is important to declare that in the analysis below, the emphasis is given to the "second" subsystem, the one that consists of the DC Bus, INV, $L_I$, TF and $C_G$. A controllable current source plays the role of the first subsystem, that drives power to the DC Bus. Generator dynamics and MPPT algorithms [27–29] were simulated but not presented, since there was no experimental confirmation and are outside the scope of this paper.

## 3. Control Architecture

The analysis of the control system starts with the dq0 transformation of sinusoidal voltages and currents, around the fundamental frequency of 50 Hz, so that PI controllers can follow the equivalent transformed DC reference signals. In addition, a three-phase PLL is implemented [30,31], in order to measure accurately the grid's frequency and phase. dq0 and PLL are entangled, since the first needs the phase angle of grid's voltage and the latter the Vq, as can be seen in Figure 3.

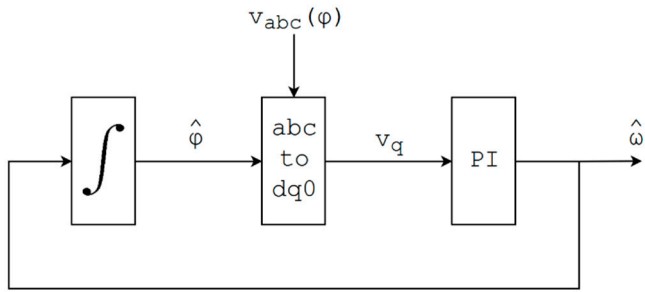

**Figure 3.** PLL Block Diagram.

Let $\hat{\varphi}$ be the estimation of phase angle $V_A$ and $\varphi$ its real value. Then, Equation (1) can be derived, in steady state:

$$\left. \begin{array}{l} u_d(t) = |V| \cos(\hat{\varphi} - \varphi) \\ u_q(t) = -|V| \sin(\hat{\varphi} - \varphi) \\ u_0(t) = 0 \end{array} \right\} \tag{1}$$

For $\hat{\varphi} \to \varphi$, $\sin(\hat{\varphi} - \varphi) = 0$, hence Vq is driven to a PI controller which, if stable, tends its input to zero. The output of the PI is grid's angular frequency and by feeding it to an integrator, grid's phase angle is derived. Finally, this is fed to the dq0 transformation block and the loop continuous. As will

be presented in Section 6, this is a simple, yet effective configuration, that is suitable for large, stable electric grids, though its response to grid faults has not been tested.

Regarding the control architecture, the first step is to formulate the state space equations. The simplified inverter-grid model is presented in Figure 4, where:

- $v_{ai}$: basic voltage harmonic of inverter's output
- $v_{ag}$: grid voltage
- L: interconnection inductor

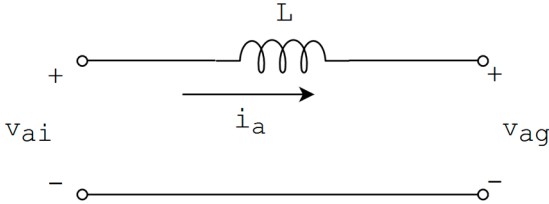

**Figure 4.** Simplified phase A inverter-grid model.

Let the grid's and inverter's voltage be:

$$\left.\begin{aligned} v_{ag} &= v_{dg}\sin(\omega t) \\ v_{ai} &= v_{di}\sin(\omega t) + v_{qi}\cos(\omega t) \\ i_a &= i_d\sin(\omega t) + i_q\cos(\omega t) \end{aligned}\right\} \tag{2}$$

Then:

$$v_{ai} = v_{ag} + L\frac{di_a}{dt} \tag{3}$$

By replacing Equations (2) to (3), Equation (4) is obtained:

$$\begin{aligned} v_{di}\sin(\omega t) + v_{qi}\cos(\omega t) &= v_{qi}\sin(\omega t) \\ &+ L\omega i_d\cos(\omega t) + L\sin(\omega t)\dot{i}_d \\ &- L\omega i_q\sin(\omega t) + L\cos(\omega t)\dot{i}_q \end{aligned} \tag{4}$$

and by solving for $\dot{i}_d$, $\dot{i}_q$, state space Equation (5) is obtained:

$$\begin{bmatrix} \dot{i}_d \\ \dot{i}_q \end{bmatrix} = \begin{bmatrix} 0 & \omega \\ -\omega & 0 \end{bmatrix}\begin{bmatrix} i_d \\ i_q \end{bmatrix} + \frac{1}{L}\begin{bmatrix} v_{di}-v_{dg} \\ v_{qi} \end{bmatrix} \tag{5}$$

where $\{i_d, i_q\}$ are the states and $\{v_{di}, v_{qi}\}$ the inputs.

To increase the current $i_d$, its derivative must be positive. This means that voltage $v_{di}$ must increase. The current $i_q$ cannot be instantly increased because of the inductor, so $i_d$ is proportional only to $v_{di}$, for the short period of time where the transition occurs. By defining the error $e_{id}= i_d^*-i_d$, it is obvious that an increase at $i_d^*$ (asterisk (*) means reference value), causes a positive error and by feeding it as input to a PI controller, it will cause the desired increase at $v_d^*$. This means that $v_{di}$ can be associated to $i_d$ and with positive PI parameters. By following the same logic, $v_{qi}$ can be associated with $i_q$, also with positive PI parameters.

At steady state, $i_d$ is proportional to $v_{qi}$ and $i_q$ to $v_{di}$, as shown in Equation (6).

$$\left.\begin{aligned} i_d &= \tfrac{1}{\omega L}v_{qi} \\ i_q &= -\tfrac{1}{\omega L}\left(v_{di}-v_{dg}\right) \end{aligned}\right\} \tag{6}$$

Active and reactive power can be calculated, at dq plane, by using the Equations (7) [32].

$$\left.\begin{array}{l} p = \frac{2}{3} v_{dg} i_d \\ q = -\frac{2}{3} v_{dg} i_q \end{array}\right\} \tag{7}$$

The active power p is directly proportional only to $i_d$ and reactive power q to $i_q$. It is obvious that for an increase in p is required to increase $i_d$. However, a desired increase in q occurs by decreasing $i_q$. The assignment of setting $\{i_d{}^*, i_q{}^*\}$ values, is made to two PIs, the outputs of which are led as reference signals in the PI group previously analyzed. PI, regulating $i_d{}^*$, has positive parameters, its output is $i_d$, while the PI regulating q has negative parameters and its output is $i_q{}^*$. Instead of the negative parameters, the error definition is reversed $e_q = q - q^*$.

Reference signal $p^*$ is calculated by the last PI. The goal of this step is to control the injected active power in the grid by stabilization of the DC Bus voltage. Figure 5 shows the schematic equivalents of IBC, DC Bus and INV. The IBC is represented as an independent current source, the value of which is determined by the MPPT. The INV is represented by a dependent current source, the value of which is set by its control, in order to maintain the DC bus voltage at 120 V.

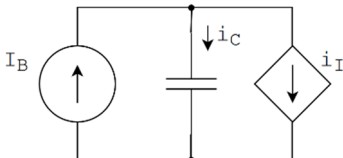

**Figure 5.** Interleaved boost converter (IBC), DC Bus and inverter (INV) equivalent model.

KCL: $i_C = i_B - i_I$ and by substituting capacitor's dynamic equation:

$$\frac{dv_C}{dt} = \frac{1}{C}(i_B - i_I) \tag{8}$$

Let $V_C > 120$ V. In that case, its derivative must become negative, in order to decrease to 120 V. So, $i_I$ must increase, hence the active power $p = V_C \cdot i_I$ will increase. This means that PI parameters must be negative, or that PI input error can be defined as $e_{V_C} = V_C - V_C^* = V_C - 120$. Finally, the whole control system is determined and shown in Figure 6. Transformation from Cartesian $\{v_d, v_q\}$ to Polar $\{m, \delta\}$, is done by Equation (9):

$$\left.\begin{array}{l} m = \sqrt{3}\frac{|V_{REF}|}{V_{DC}} = \sqrt{3}\frac{\sqrt{v_d^2 + v_q^2}}{V_{DC}} \\ \delta = \arctan\left(\frac{v_d}{v_q}\right) \end{array}\right\} \tag{9}$$

where $V_{DC} = const. = 120$ V

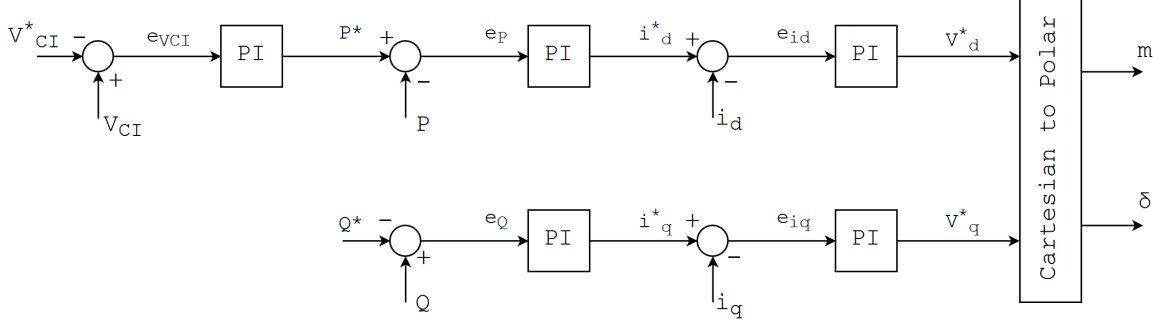

**Figure 6.** Control system.

Reference signal for the reactive power q is given by the user, but it can also be automatically generated by measuring the voltage of grid's bus. This was pointless in this particular case, because the system under study cannot provide enough q to significantly affect the above voltage. While p takes values from 0 to 1000 W, q is limited from 0 to 800 VAR.

## 4. Investigation of PI Parameters

Finding the PI parameters can be done by modeling and linearizing the system around the operating point, or empirically by adjusting the parameters until the system has the desired response. The first method requires more time and real-time linearization (extra computational cost), as the system does not have a single point of operation, but it guarantees stability. The second does not requires a model, it is implemented quickly, however it cannot mathematically guarantee stability, but only through specific state transitions. The second method was preferred, which is a modified version of Ziegler-Nichols [33,34], so it can be applied to nonlinear systems, as the one we are dealing with. The tuning of the PI controllers was done by following the steps below:

1. Starting from an arbitrary value of $K_P$ and for $K_I = 0$. Based on the above analysis, $K_P$ is positive and it affects the speed at which PI output reaches the reference signal, something that is well known from control theory.
2. A random series of extreme transitions is assigned to the system. If the system is unstable or the variables are congruent with the constraints, at any time, $K_P$ is reduced. Otherwise, $K_P$ is increased. Extreme transitions are those between extreme states or points of operations, which in this case are {p, q} = {0, 0}, {1000, 0}, {0, 800}, {1000, 800} {[W], [VAR]}.
3. Step 2 is repeated until the system is close to the marginal stability or limitations. Let be the critical gain $K_u = K_P$ for this situation. In addition, for this $K_P$ the $T_u$ of the oscillation of the variables are calculated.
4. From the equations of Table 1 (Ziegler-Nichols Method), the values of $K_P$ and $K_I$ are calculated, according to the control method. In this case PI controller is used, since PID was vulnerable to instability.
5. Based on the values of Step 4, small changes are made to optimize the response. Typically, $\pm 15\%$ of the original calculated value.

**Table 1.** PI parameters calculation, according to Ziegler-Nichols Method [33,34].

| Controller | $K_P$ | $K_I$ | $K_D$ |
|:---:|:---:|:---:|:---:|
| P | 0.5 Ku | - | - |
| PI | 0.45 Ku | Tu/1.2 | - |
| PD | 0.8 Ku | - | Tu/8 |
| PID | 0.6 Ku | Tu/2 | Tu/8 |

Since the system is nonlinear and no mathematical model is used, it is impossible to prove that the result of this procedure is optimal. According to the simulations and the physical experiments, though, the response time is less than 300 ms, which is totally acceptable.

The above described procedure is painful for the designer if it is made manually, however it can be fully automated by creating a program that follows the steps. This has the advantage of not only avoiding the repetitive work of fine tuning, but also test the stability and system response for a wider range of operating point transitions.

## 5. DC Bus Auto-Charging and Reactive Power Injection

Proposed design of Figure 2, combined with the control architecture of Figure 6, offers an additional function, as it can charge the DC Bus, regardless of the wind conditions, hence providing

reactive power injection. Let DC Bus be uncharged or charged below reference voltage of 120 V. When the system is connected to the grid and the control system is active, the error $e_{VCI}$ will become negative and therefore P* will become also negative, hence the system will drain active power from the grid until DC Bus voltage reaches 120 V.

When the wind speed is low, the wind turbine cannot generate any significant current and $I_B$, from Figure 5, is zero. Therefore, Equation (10) gives the DC Bus voltage, as a state, with input variable the inverter's current $i_I$.

$$\left. \begin{array}{l} i_C = -i_I = C\frac{dV_C}{dt} \\ V_C = -\frac{1}{C}\int i_I\, dt \end{array} \right\} \Rightarrow \dot{V}_C = -\frac{1}{C}i_I \tag{10}$$

Charging can be divided in to the following phases:

- When the voltage of DC Bus is less than ~15V, antiparallel diodes are forward biased and start conducting, charging the bus.
- After that, inverter can modulate its voltage output, so that power flows from grid to the bus, by utilizing a negative δ value, hence a negative P value.

After charging is complete, the system can inject reactive power to the grid. Since current is non-zero, the system draws a small amount of active power from the grid, as solid-state switches and copper conductors present losses.

All the above add a new functionality, as the RES system can provide essential aid to the main power system, especially if the latter is an autonomous–non-interconnected, even if the environmental conditions do not permit active power generation by the RES. This is making RES systems more attractive and has multiple advantages for the power grid:

- Increased stability and better transient behavior
- Reduced transmission losses
- Reduced emissions of greenhouse gases

## 6. System Simulation and Experimental Results

In order to verify the proper operation of the second subsystem, two steps were taken. First, the power circuit was simulated with open-loop (OL) control, to verify the theoretical behavior that was expected from the analysis. Second, the closed-loop (CL) control, shown in Figure 6 was introduced, one layer at a time, starting from right to left. Each layer had its parameters configured with the procedure descripted previously. The active power reference is set by the inrush current, as shown in Figure 7. When the system's response was the desired one, the next layer was added. By the end of this process, the system was able to respond quickly to any reference condition, as is shown in Figures 8–10, which is divided in the following time periods:

- t = [0, 0.1] sec. control is offline and the PLL is synchronized with the grid's frequency and phase.
- t = [0.1, 0.6] sec. the DC Bus is charging from 0 to 120 V. The overshoot is less that 135 V, which is acceptable since the capacitors of the DC Bus were designed for 200 V.
- t = [0.6, 1.5] sec. both active and reactive power are set to zero. Since voltage of the DC Bus is considered constant at 120 V, input active power is defined only by the current of CCS (independent current source).
- t = [1.5, 3.5] sec. active power is set to 1000 W (or CCS reference current is set to 8.3 A) and reactive power is set to 800 VAR.
- t > 3.5 sec. active power is set to 0 W and reactive to 800 VAR.

At each transition, voltage of the DC Bus deviates, but returns to the operating point in less than 300 ms. Active and reactive power may surpass the maximum values of {1000 W, 800 VAR} respectively, but this is for short period of time and unable to harm any component.

System was simulated by dividing it to two subsystems, with separating point that of the DC Bus, as shown in Figure 3. Figure 5 presents the analytical model of the dependent current source, that is, the inverter, which regulates the power flow to the grid by stabilizing voltage of the DC Bus at 120 V. The first subsystem, that is, the PSMG and the IBC was simulated and tested with several MPPT algorithms. The reason of this separation is the computational cost and the time that is necessary for the simulation and is justified by the large capacity of the DC Bus.

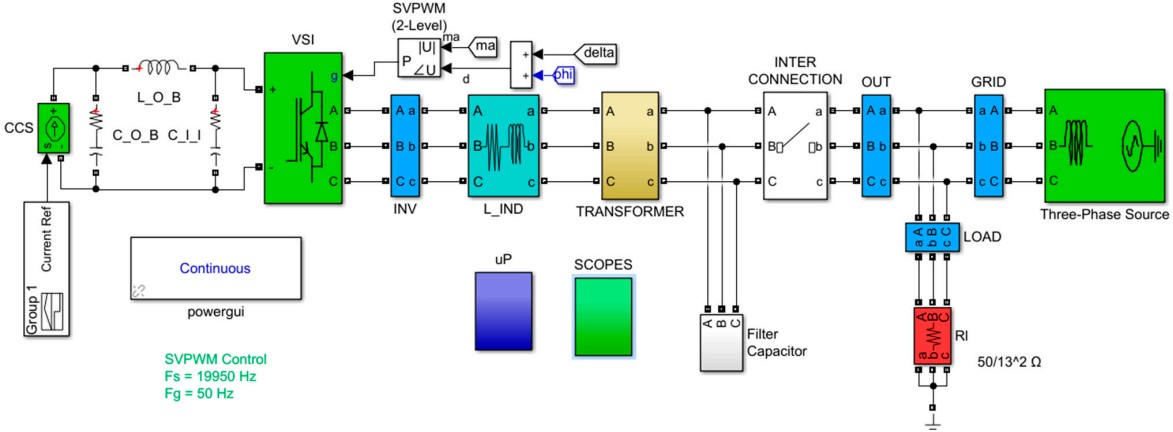

**Figure 7.** Simulation of the power circuit.

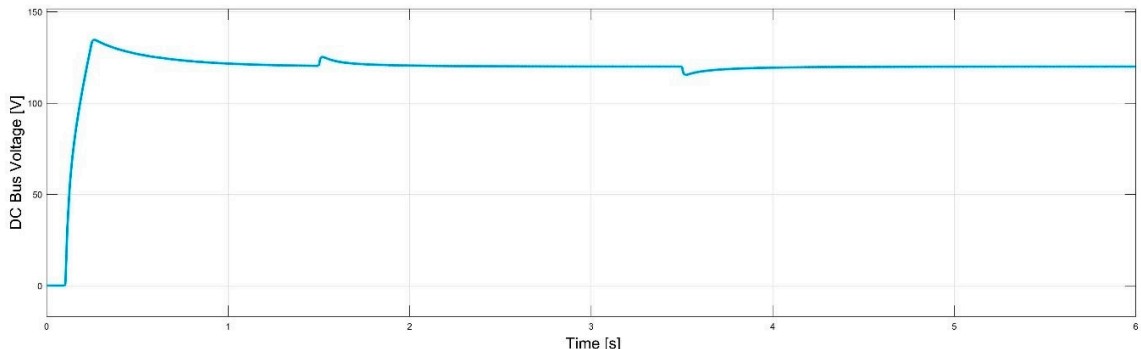

**Figure 8.** Charging, transition and steady-state voltage of DC Bus, {P, Q} = {0, 0}, {1000, 800}, {0, 800} {[W], [VAR]}.

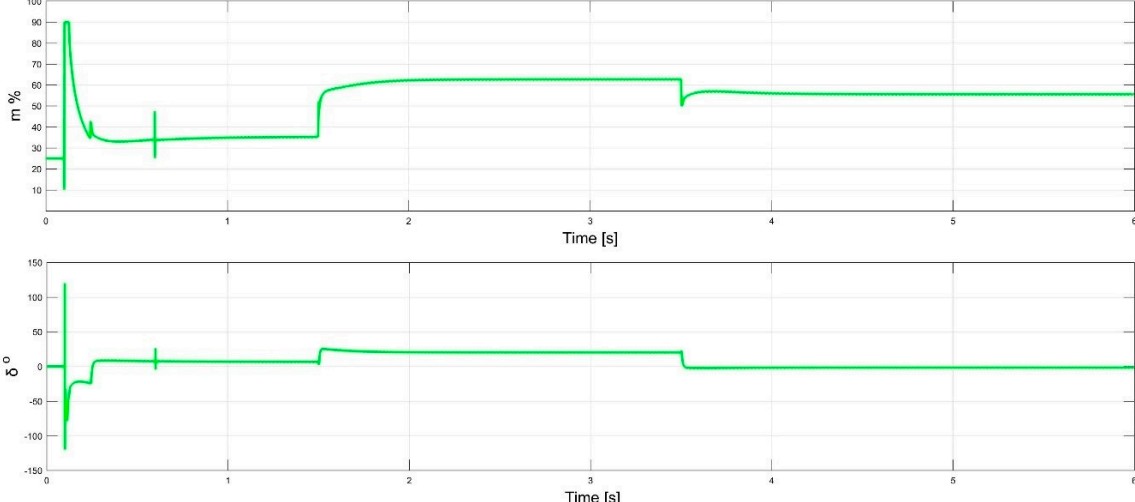

**Figure 9.** Charging, transition and steady-state voltage of m and δ, {P, Q} = {0, 0}, {1000, 800}, {0, 800} {[W], [VAR]}.

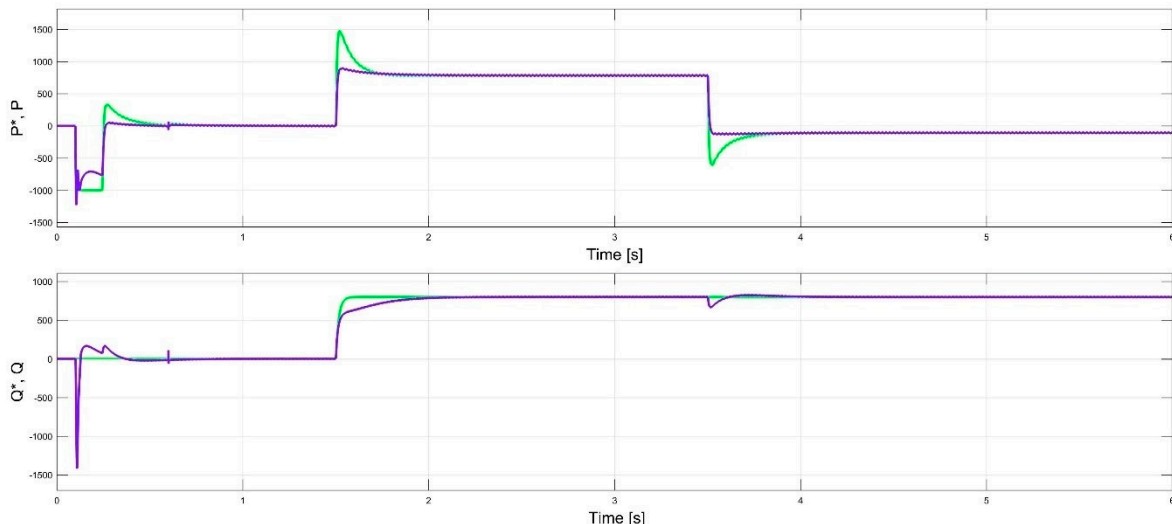

**Figure 10.** Charging, transition and steady-state voltage of P and Q, {P, Q} = {0, 0}, {1000, 800}, {0, 800} {[W], [VAR]}.

As the control of each simulation has taken its final form and the parameters were configured by the method described above, the final step is the digitization of the "analog" measurements, PIs, PLLs, etc.

All the necessary boards were designed and populated, as shown in Figure 11, which presents the second subsystem, which consists of the DC Bus, the inverter, the interconnection coil, the transformer and the current filtering capacitors, but also the measurement and filtering boards and the control board of the inverter. Figure 12 presents each individual layout of the power board of the inverter.

While the programming of the microcontroller that was used (STM32F429ZI) was far from trivial, by following the logic structure of simulation and using the PI parameters found, system managed to work as previously described. The role of the independent current source was performed by a DC power supply in constant current operation, while reactive power reference value was set directly by a potentiometer.

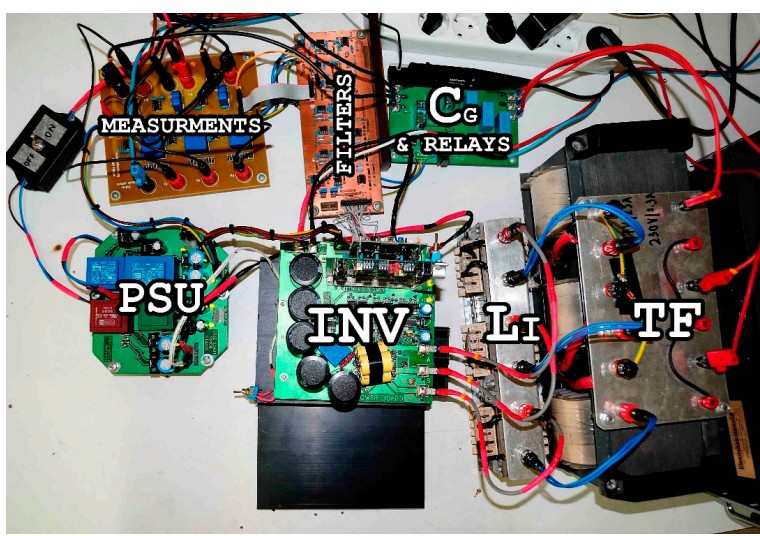

**Figure 11.** Complete physical subsystem.

The last step, before connecting to the grid, was to confirm the proper operation of the PLL. This was done by driving the phase angle $\hat{\varphi}$ to a DAC port of the microcontroller and display it in the same window with grid's voltage. This is shown in Figure 13. PLL synchronization was achieved

in approximately 60 ms. A small disturbance can be seen, as a result of lab's transformer saturation. After closing the interconnection switch and in OL control, various test were made, one of which is presented in Figure 14.

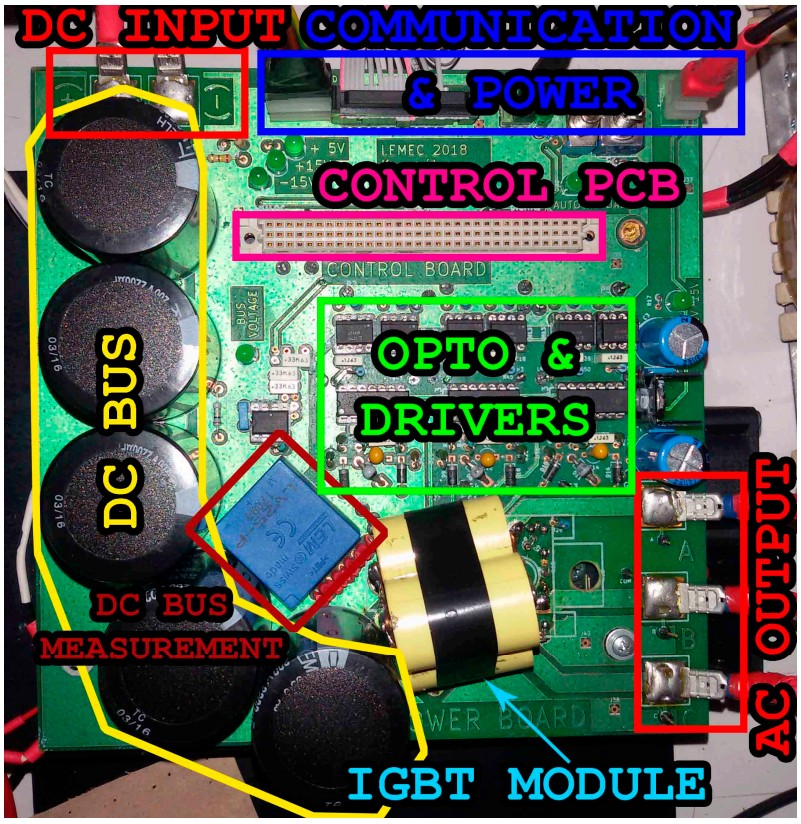

**Figure 12.** Power board—Inverter.

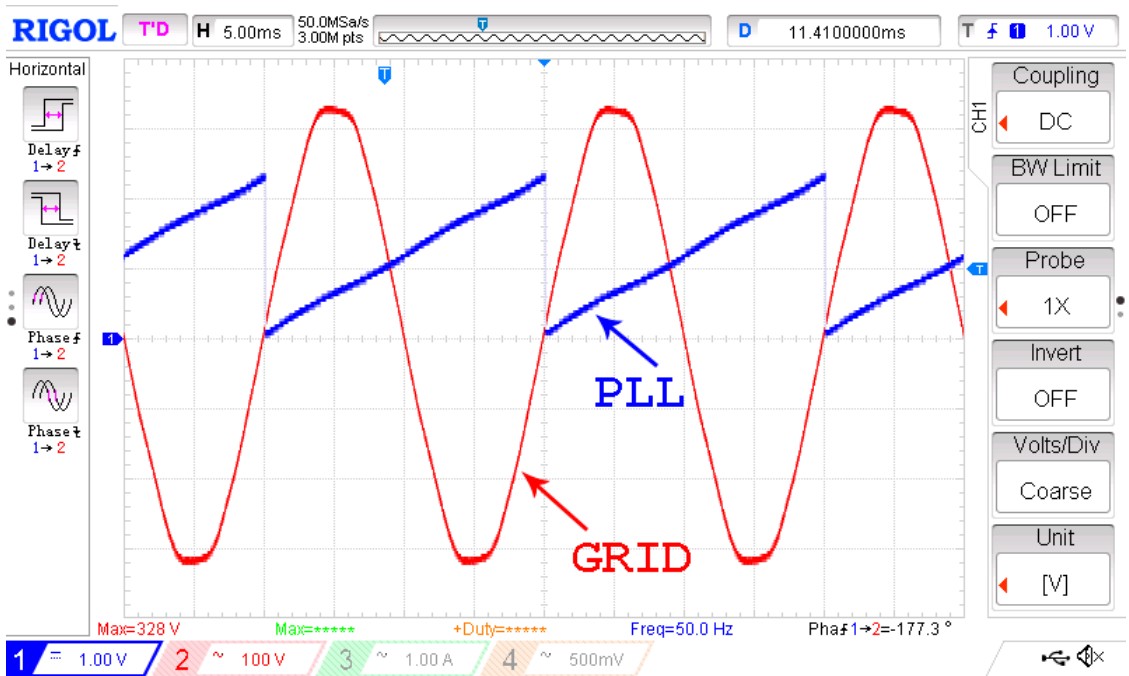

**Figure 13.** Grid's phase A and PLL's output.

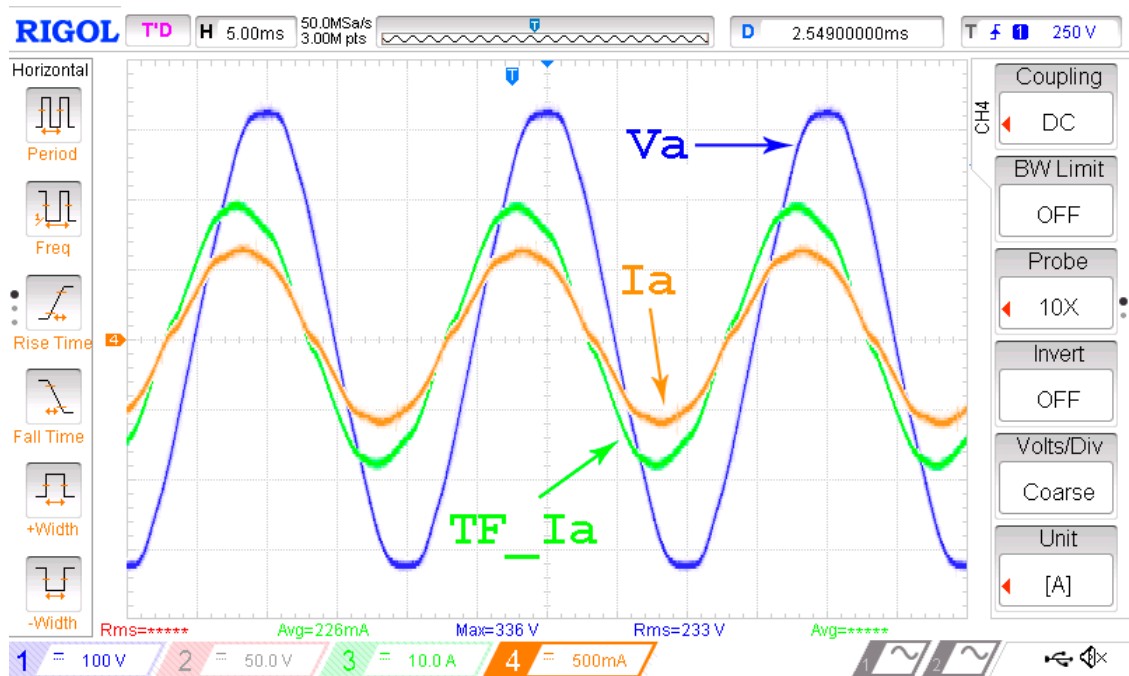

**Figure 14.** Grid's phase A voltage, phase A current and transformer's primary current, P = 350 W, Q = -100 VAR.

After verifying the proper operation of OL control, CL control was applied. Figure 15 presents the grid's voltage, input current (grid's injected current) and instantaneous power, for maximum input power of 1000 W. Due to power losses over parasitic resistances of inductor and transformer, system performance was 76%. Figure 16 presents the same values, but for active and reactive power of 500W and 500 VAR respectively, measured at the output, that is, the gird. These figures justify the proper operation of both the power and the control system.

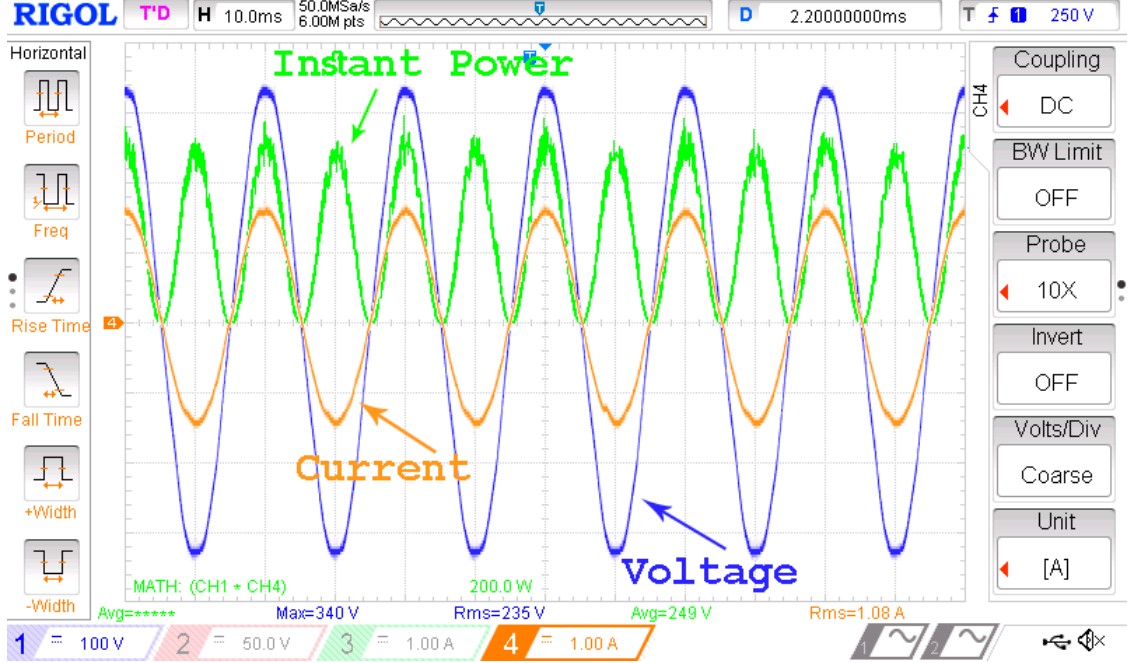

**Figure 15.** Voltage and current, P = 750 W, Q = 0 VAR.

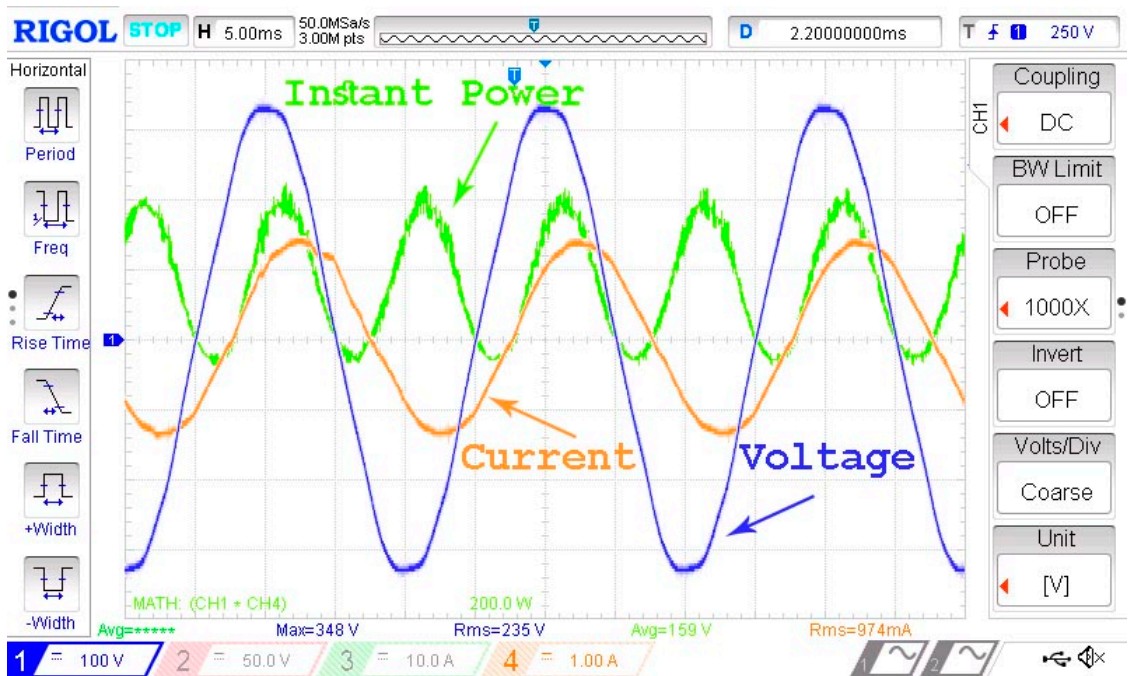

**Figure 16.** Voltage, current and instant power, P = 500 W, Q = 500 VAR.

The experimental configuration of the second subsystem, where the WT and the IBC are substituted by a DC power supply, can be seen in Figure 17. Power from the DC supply is at maximum (120 V × 8.3 A = 1 kW), hence the system is at full power too.

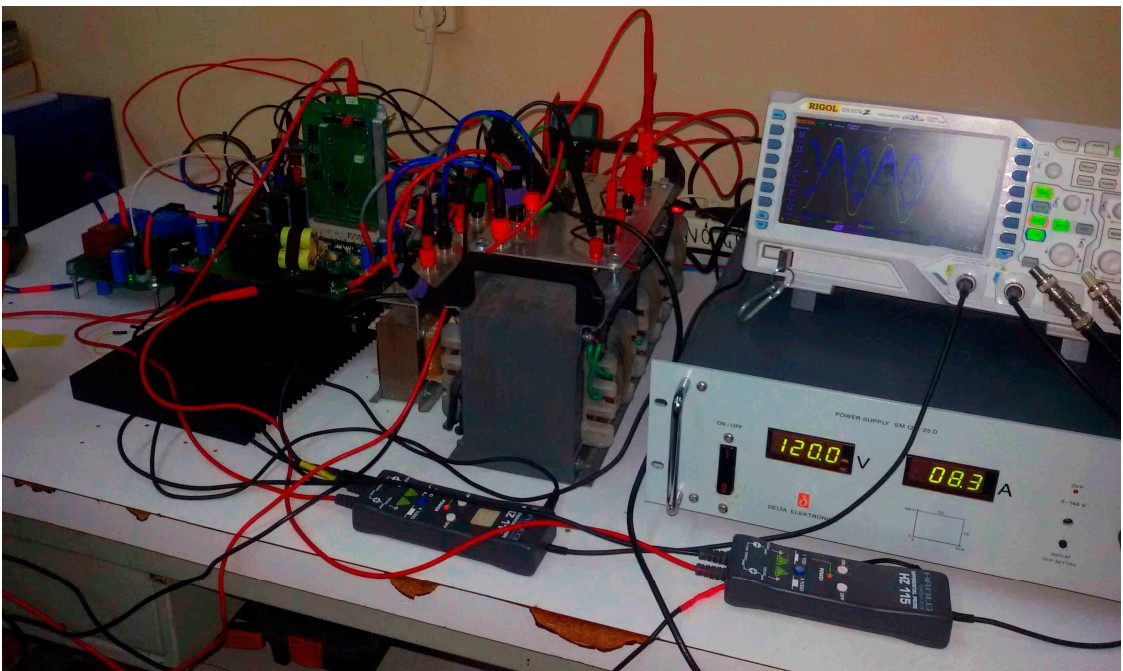

**Figure 17.** DC power supply in constant current mode at full power–1 kW and CL control.

## 7. Conclusions

In this paper, is developed a method of interconnecting a DC source, such as a PMSG with a FBR or a DC power supply, with the low voltage distribution grid, via an inverter, which is responsible for regulating active and reactive power. At the same time, the whole system was designed in order

to present as many components, machines and converters as possible, in order to fulfill the teaching purposes of LEMEC and diploma thesis. In addition, an improved control method and architecture is introduced, so it can be easily configured to any similar system and to respond fast to the wind changes.

In order to safely confirm the operation of both the power and control modules, a model of the system has been developed, based on Matlab/Simulink simulation program. Extreme transitions were set, to confirm the stability and the response time of every component and have a crude estimation of all currents and voltages.

The result of this research is a stable and reliable working system, with well documented components, many on-board testing points and a confirmation of the ability to configure real PI controllers from a model simulation. Additionally, the research proved that it is possible to charge the DC Bus from the power grid and use the inverter subsystem as reactive power compensator. All of the above are very useful, especially today, when RES are starting to gain a respected piece of the power generation market, which targets to a more environmentally friendly operation.

**Author Contributions:** The above research was done by T.P. and E.K., for their diploma thesis, with original title "Study, Construction and Control of a Wind Turbine System for Interconnection with the Low-Voltage Grid", UoP, Patras, July 2018. System investigation and analysis, hardware design and construction and software programming were done by T.P. and E.K. Supervision of the whole research was done by E.T. Writing of this paper was done by T.P, with crucial contributions and comments from E.T.

**Funding:** This research received no external funding.

**Conflicts of Interest:** The authors declare no conflict of interest.

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
