# Peer review of "Improved Active and Reactive Control of a Small Wind Turbine System Connected to the Grid"

_resources, doi:10.3390/resources8010054_

Round 1
Reviewer 1 Report
The method of control must be presented in a more detailed manner i.e the PLL description and the MPPT algorithm or equations used are missing. The method of PI regulators parameter identification is trivial. plese consider to extent the number of references (IEEE transactions rule of thumb nr of pages x 3 citations = 36 citations)
Author Response
Dear Reviewer,
Thank you very much for your comments on our paper. Your suggestions were highly appreciated and helped us a lot to improve the structure of it. We think that all of your suggestions have been fulfilled in this new version. Following, specific answers are included to each of your suggestions.
All corrections are highlighted with yellow color in the revised manuscript, except from some minor changes in language.
Introduction has been extended and includes plenty of references. We add two paragraphs presenting relevant work done from “power system” and “control system” point of view, respectively. Regarding the first one, there is not much to comment on, since the theory behind is well established from any power systems textbook. For the second one, there are some control architectures suggested in references. We utilize one that provides control on pq power, idq currents and vdq voltages reference signals and apply the proper restrictions, while at the same time the response is quick enough and within boundaries. Also, in section 3 (Control Architecture) we present another way for decoupling active and reactive power.
1. The method of control must be presented in a more detailed manner i.e the PLL description and the MPPT algorithm or equations used are missing
Answer: We add a brief analysis of the PLL presented in section 3 and we add an oscilloscope waveform in section 6. MPPT algorithms were not presented on purpose, since our paper emphasizes on the second subsystem (the one with the inverter, right of the DC Bus). The first subsystem (the wind turbine with the IBC) was simulated but not experimentally tested, so it was left out. Its role was played by a DC power supply, in constant current mode.
2. The method of PI regulators parameter identification is trivial.
Answer: When we were trying to tune the PI controllers, we did not come across any suggestion of how this can be achieved on a non-linear system, without a detailed model. The method we present can be seen as a modified Ziegler – Nichols method and we believe that it can help any control designer, since our paper proofs that it works, with experimental confirmation.
3. Please consider to extent the number of references (IEEE transactions rule of thumb nr of pages x 3 citations = 36 citations)
Answer: We extend the number of relevant references, most of which are conference papers found in IEEE site.
Reviewer 2 Report
It's a good example for how works a small eolic plant to your students, Maybe will be a good idea to enlarge the theoretical description of how works and include more experimental results regarding the reactive control.
Author Response
Dear Reviewer,
Thank you very much for your comments on our paper. Your suggestions were highly appreciated and helped us a lot to improve the structure of it. We think that all of your suggestions have been fulfilled in this new version. Following, specific answers are included to each of your suggestions.
All corrections are highlighted with yellow color in the revised manuscript, except from some minor changes in language.
1. Maybe will be a good idea to enlarge the theoretical description of how works
Answer: In section 1, we add a figure which explains the use of each device. Also, in section 3, we add a brief explanation of the PLL we used and, in section 6, some debugging methods and observations..
2. Include more experimental results regarding the reactive control.
Answer: In section 6 we add a figure that presents the operation of the system in OL control. Reactive power in this case is negative, since angle φ is negative..
Reviewer 3 Report
Dear Authors,
You have made a very good work. Your contribution has a logical structure and designed appropriately. Research question clearly outlined and data presented in an appropriate way. Conclusions answer the aims of the study and supported by references or results.
The only drawback of the paper is references, 4 of 12 are not relevant (years 1997, 1998 and 1942), please try to provide more recent references. It is good to make more recent review in this field. Moreover, references [6]-[8] does not have publication year.
Author Response
Dear Reviewer,
Thank you very much for your comments on our paper. Your suggestions were highly appreciated and helped us a lot to improve the structure of it. We think that all of your suggestions have been fulfilled in this new version. Following, specific answers are included to each of your suggestions.
All corrections are highlighted with yellow color in the revised manuscript, except from some minor changes in language.
1. references, 4 of 12 are not relevant (years 1997, 1998 and 1942), please try to provide more recent references. It is good to make more recent review in this field.
Answer: We extended the number of relevant references, most of which are conference papers found in IEEE site. The reference of 1942 is the famous article of J.G. Ziegler and N.B. Nichols. We also add a reference to the great book of K. Ogata, which presents many examples of ZN method. Furthermore, we extended the introduction, so it includes the most recent works in the field of interconnection systems.
2. Moreover, references [6]-[8] does not have publication year.
Answer: Fixed.